# Willingness to use electronic medical record (EMR) system and its associated factors among health professionals working in Amhara region Private Hospitals 2021, Ethiopia

**Andualem Fentahun Senishaw**[1]*, **Biniyam Chakilu Tilahun**[2‡], **Araya Mesfin Nigatu**[2‡], **Shegaw Anagaw Mengiste**[3‡], **Karen Standal**[3‡]

1 Department of Health Informatics, College of Health Sciences, Debre Markos University, Debre Markos, Ethiopia, 2 Department of Health Informatics, Institute of Public Health, College of Medicine and Health Science, University of Gondar, Gondar, Ethiopia, 3 School of Business, Institute of Business, History and Social Science, University of South-Eastern Norway, Notodden, Norway

‡ BCT, AMN, SAM and KS are a second equal contributions to this work.
* andugissa@gmail.com

**Data Availability Statement:** All relevant data are within the paper and its Supporting Information files.

## Abstract

### Introduction

Despite the high expectations of electronic medical records as a great prospect for improving performance in healthcare, the level of adoption and utilization, particularly in a developing country, is low. Knowing the willingness to use the electronic medical record system in the private hospital has an impact on the future implementation status and utilization of the electronic medical record in Ethiopia. However, there was no evidence of the status of the willingness to use electronic medical record systems in private hospitals in the Amhara region. This study aimed to assess the willingness to use electronic medical record Systems and its associated factors among health professionals working in Amhara Region Private Hospitals.

### Methods

A cross-sectional institutional study was performed among 406 health professionals selected using proportional allocation with a simple random sampling technique in Amhara region private hospitals by using self-administered structured questionnaires. The data were analyzed using SPSS version 20 software. Descriptive statistics and binary logistic regression were performed to estimate the crude and adjusted odds ratios with a 95% Confidence interval.

### Results

Out of the 406 participants included in the analysis, 307 (75.6%) showed a willingness to use the electronic medical record system. About three hundred twelve (76.8%) health

**Funding:** The author(s) received no specific funding for this work.

**Competing interests:** The authors have declared that no competing interests exist.

professionals had good knowledge of electronic medical record systems, and 257 (63.3%) had good computer skills in electronic medical record systems. Health professionals who had electronic medical record knowledge (AOR = 1.85, 95% CI (1.004–3.409)), EMR training (3.29, 95% CI (1.353–8.003)), technical support personnel (1.92, 95% CI (1.122–3.305)), supportive supervision (AOR = 1.97, 95% CI (1.072–3.628)), and computer skill on electronic medical record (1.77, 95% CI (1.002–3.148)) were significantly associated with the outcome variable.

## Conclusions

This finding shows a good proportion of willingness to use the electronic medical record system. The most significant factors associated with willingness to use the electronic medical record system were a lack of computer skills, computer training, and knowledge of the electronic medical record system.

## Introduction

All over the world, health information systems (HIS) and technologies are being utilized more frequently and are viewed as useful approaches to improving the effectiveness of the health care system and the standard of patient care [1]. Healthcare organizations employ HIS like electronic medical records (EMR), telehealth, mobile health, and health information management systems to achieve these results [2].

The national-wide e-health approach toolkit developed through the World Health Organization (WHO) and International Telecommunication Union (ITU) [3] defines EMR as "a computerized medical record used to capture, store, and share information among healthcare providers in an organization, supporting the delivery of health services to patients."

Even though there may be an excessive expectation and need for EMR as a great prospect for improving quality, continuity, and performance in healthcare, the general adoption status, particularly in a developing country, is low [4]. And also, the status of EMR implementation in the Amhara region private hospital is not uniform throughout all the departments. Many reasons can be given for the low utilization and adoption of EMR in Ethiopia, but the main ones are related to the skill gap, lack of knowledge, lack of availability of technical support and infrastructure, perception, willingness, technology resistance, and administrative problems that have slowed the process of implementation and adoption of EMR [5–7].

Knowing the willingness to use the EMR system in private and public hospitals has an impact on the future implementation status and utilization of the EMR system in Ethiopia [8]. Particularly this study focuses on the private health facilities since the attention was not given to the private hospitals towards the willingness of EMR system with scientific evidence. The Ethiopian government needs to digitalize public as well as private health facilities and needs to strengthen the public-private partnership. Public-private partnerships and private sector participation are encouraged by the overall policy of the Ethiopian health system [9, 10].

In the Ethiopian health system, the majority of private health facilities are found in urban areas, and the majority of the population seeks health services from private health facilities. The private experience of implementation and utilization of the EMR system appears slightly different from that of public health agencies, as most paramedics and other care providers are highly dependent on payments for what is recorded in the EMR system, leading to a relatively

better status of EMR implementation. However, the focus on collecting a sufficient amount of structured and unstructured patient data in every encounter is highly limited to certain parameters [11].

Ethiopian health policies and strategies have largely ignored the attitude and behavioral factors (willingness) of healthcare professionals in both private and public hospitals to use the EMR system, focusing instead on the material aspect, specifically the installation and infrastructure of EMR systems [8].

To overcome the issues related to EMR and to facilitate the adoption and use of innovative technologies, it is very important to understand the factors that affect the willingness to use the EMR system by healthcare professionals. Health professionals' Understanding, knowledge, willingness, and mindset of health professionals regarding EMR use are critical factors that may influence its future fulfillment [6].

Studies have shown that a lack of knowledge, skills, availability of skilled human capital, resource availability, computer literacy, English language proficiency, educational status, and training are the factors that potentially affect the willingness to use EMR systems by healthcare professionals [8]. Most of the problems are related to the unwillingness of its users to accept the transition from paper-based to electronic systems. This has contributed to the failure costs of many EMR installations [4].

Having electronic medical records systems in many health organizations improves the efficiency of health care delivery, ensures confidentiality, improves the overall healthcare system, and increases the availability of health information, and also retrieves patient information for patient care, statistics, research, and teaching [6, 12, 13]. Therefore, adopting EMR in a private health facility is one of the digital technologies that can successfully transform the health system by reducing the burden on government health facilities while making the service fast and easy in the private sector.

There has been no study conducted in the Ethiopian context on the willingness to use EMR among health professionals working in private hospitals. Therefore, the broad objective of this study is to assess the willingness to use electronic medical record (EMR) systems and its associated factors among health professionals working in Amhara Region private hospitals. The specific objective is to determine health professionals' willingness to use the EMR system and to identify factors associated with the willingness to use the EMR system in Amhara region private hospitals in 2021.

## Materials and methods

### Study design and setting

A cross-sectional institutional study was performed from January 25 to February 20, 2021. The study was conducted in private hospitals found in the Amhara region. The Amhara Region is located in the northwestern part of Ethiopia, and the region is divided into 13 zones and 140 weredas. This region has a total of 4490 public health facilities (81 hospitals, 859 health centers, and 3550 health posts) and 1165 private health facilities (1157 clinics and 10 hospitals). The distributions of these 10 hospitals are four from Bahirdar town woreda HO, three from Dessie town woreda HO, two from north Shewa ZHD, and one from Gondar town woreda HO. These private hospitals are Yifat Hospital, Afilas Hospital, Salam Hospital, Adina's Hospital, Ayu Hospital, Bati Hospital, Dream Care Hospital, Ethio Hospital, Gamby Hospital, and IBEX Hospital.

### Sample size determinations, study participants, and sampling procedures

The sample size of this study was determined using a single population proportion formula, considering the following assumptions: With a 95% level of confidence, a 5% margin of error,

and a 10% non-response rate, a proportion (p) of 50% of those willing to use the EMR system was calculated. In the end, 423 people were included in the sample. The source population was all health professionals working in private hospitals in the Amhara region, and the study population was selected health professionals who had been working in ten private hospitals in the Amhara region. Health professionals who were working for less than six months and who were working both at private and public health facilities were excluded to avoid bias.

There were ten private hospitals in the Amhara region. From each hospital, permanent health professionals working during the study period were included in the study. For each hospital, a proportional allocation was made based on the actual number of health professionals. Each healthcare worker in each private hospital was selected by simple random sampling from a list of medical staff administrative records.

## Operational definition

Knowledge: Knowledge was measured on a 5-point Likert scale ranging from "strongly disagree" (score 1) to "strongly agree" (score 5). Thirteen knowledge questions were added and divided by 13 to create a composite variable scale (ranging from score 1 to 5) for data analysis. Finally, the composite variable score was dichotomized as "good knowledge" or "poor knowledge" based on the final score. Accordingly, a final score of the above three (agree and strongly agree) was categorized as "good knowledge," while those final scores of three or below (strongly disagree, disagree, and neutral) were categorized as "poor knowledge" [14].

Willingness to use: The willingness of health professionals to use the EMR system was rated as "willing" or "not willing" using composite scores obtained from all five willingness questions [8]. Five questions were scored, and the maximum score obtainable is five marks. A score of 3 or above out of 5 marks suggests willingness, while a score of less than 3 marks suggests an unwillingness to use the EMR system [8].

Computer skill: Skill was measured on a 5-point Likert scale ranging from "strongly disagree" (score 1) to "strongly agree" (score 5). Each skill question was added and divided by the number of questions to create a composite variable scale (ranging from score 1 to 5) for data analysis. Accordingly, a final score above three (agree and strongly agree) was categorized as "good skill" while those final scores of three or below (strongly disagree, disagree, and neutral) were categorized as "poor skill" [14].

Technical factors: Which are technical and knowledge-related factors for using EMR in private hospitals, such as computer skills, knowledge of EMR, language difficulty, training, and using a computer for EMR.

## Data collection tools and procedures

Self-administered structured questionnaires were used to gather quantitative data by adapting questionnaires from [8, 15]. he outcome variable includes five questions with "yes" or "no" responses, as well as some multiple-choice questions. Socio-demographic, technical, organizational, and resource-related variables were included in the questionnaire. A questionnaire was prepared in English because the study participants were well educated and could easily understand English; if the questioner translated it into the local language (Amharic), it might create some difficulty to understand. Health information technicians (HIT) who have good communication skills were recruited to collect questions. Health officers who were experienced in research work supervised the data collection process. A two-day training was given for facilitators and supervisors on the objective of the study, data collection procedures, data collection tools, respondents' approaches, data confidentiality, and respondents right before the data collection date. The completeness of the questionnaires, the overall quality of the data collected,

and the daily status of the data collector were checked every day by the supervisors and investigators. Before the actual data collection, pretesting of the questionnaire was conducted outside the study area but with similar characteristics to the actual study area.

### Data management, analysis, and quality assurance

Data was entered using Epi-data version 4.6 and analyzed using Statistical Package for Social Science (SPSS) version 20. Descriptive analyses were computed for all variables in the study. Bivariable and multivariable analysis with an adjusted odds ratio was used to measure the association of dependent and independent variables, and 95% confidence intervals and P-value were calculated to evaluate statistical significance. Variables with a p-value of $\leq 0.2$ on the binary logistic regression analysis were entered and further computed on the multivariable logistic regression model. Adjusted odds ratios and 95% confidence intervals were calculated for each of the independent variables in logistic regression models with the health professional's willingness to use the EMR system as a dependent variable.

Based on the pretest, the internal consistency of each dimension of the composite variable questionnaire was checked using Cronbach's alpha. Score on EMR knowledge (Cronbach alpha = 0.79), computer skill (Cronbach alpha = 0.91), and willingness to use the EMR system (Cronbach alpha = 0.81). Based on this, they were acceptable. The goodness of fit of the model was assessed using Hosmer Lemeshow's statistical test, and its values above 5%, which is 0.76, indicate that the model has a good predictive ability. A multicollinearity test was performed for the variables included in the final multivariable model. Hence, the variables had a VIF value of less than five.

### Ethical approval and consent to participate

The study protocol was reviewed and obtained from the ethical review committee of the University of Gondar College of Medicine and Health Science, Institute of Public Health. A supporting letter was also obtained from the Amhara Regional Health Bureau. Informed written consent was obtained from each study participant after telling them the objective of the study. They were also informed about the benefits of the study. The data collection was anonymous, and the information was kept confidential.

## Results

### Socio-demographic factors

A total of 424 health professionals were recruited, and 406 participants were included in the study, with a response rate of 96%. The mean age of respondents with the standard division was 29.6±6.5 years. The age group of the respondents ranges from 20–65 years, and almost half of them 209 (51.5%) were in the age range of 20–29 years. Regarding gender, 213 (52.5%) were male respondents, and about 225 (55.4%) were single. 206 (50.7%) of the participants were first-degree holders, and 306 (75.4%) of them had working experience of less than 7 years. The majority of the respondents, 119 (29.3%), were nurses by profession, and 150 (36.9%) of the participants' monthly income ranged from 5251 to 7800 Birr (Table 1).

### Technical factors for willingness to use the EMR system

The majority 312 (76.8%) of health professionals had good knowledge of EMR system use. More than half 257 (63.3%) of the respondents had good skills to use computer systems. Almost half of 190 (46.8%) of the respondents did not receive EMR system training because of a lack of access to training 105 (55.6%), absence of interest in training 58 (30.5%), lack of time

**Table 1. Sociodemographic factors for willingness to use an EMR system among health professionals working in Amhara region private hospitals in 2021 (N = 406).**

| variable | category | Frequency | percent |
|---|---|---|---|
| Sex | male | 213 | 52.5 |
|  | female | 193 | 47.5 |
| Age in years | 20–29 | 209 | 51.5 |
|  | 30–39 | 164 | 40.4 |
|  | Above 40 | 33 | 8.1 |
| Work experience in the year | Less than 7 years | 306 | 75.4 |
|  | Above 7 years | 100 | 24.6 |
| religion | Orthodox |  |  |
|  | Christian | 315 | 77.6 |
|  | Muslim | 71 | 17.5 |
|  | other | 20 | 4.9 |
| Marital status | Single | 225 | 55.4 |
|  | Married | 166 | 40.9 |
|  | Divorced | 11 | 2.7 |
|  | Widowed | 4 | 1.0 |
| Educational level | Diploma | 142 | 35.0 |
|  | Degree | 206 | 50.7 |
|  | Master and above | 58 | 14.3 |
| Professional category | Physician | 38 | 9.4 |
|  | Nurse | 119 | 29.3 |
|  | Laboratory | 82 | 20.2 |
|  | Pharmacy | 71 | 17.5 |
|  | Midwifery | 46 | 11.3 |
|  | Others** | 50 | 12.3 |
| Monthly income in Birr | Less than 3200 | 22 | 5.4 |
|  | 3201–5250 | 145 | 35.7 |
|  | 5251–7800 | 150 | 36.9 |
|  | 7801–10900 | 51 | 12.6 |
|  | Above 10901 | 38 | 9.4 |

** Others include public health, psychiatry, optometry, radiology, physiotherapy, and dental.

to be self-trained 57 (30%) and 35 (18.4%) health professionals said their work din not require training. About two-thirds, 261 (64.3%) of health professionals had no English language barrier to using a computer and an EMR system. 245 (60.3%) of the respondents used a computer device in the EMR system for the following obvious purposes: writing reports (33.5%), lessening music (16.7%), reading (29.8%), and keeping patient files (53.9%), (Table 2). The main reason for not taking EMR training in Amhara region private hospitals was not having access to EMR system training.

## Organizational and resource-related factors

Of the total respondents, 230 (56.7%), 225 (55.4%), and 183 (45.1%) were able to access computers, the EMR guidelines, and the Internet to run an EMR system, respectively. More than half of the 212 (52.2%) healthcare professionals were assisted by trained IT technical personnel hired for EMR system maintenance. Nearly half 210 (49.0%) of respondents got managerial support to use the EMR system; however, only 158 (38.9%) of the respondents said that an

**Table 2. Technical factors for willingness to use an EMR system among health professionals working in Amhara region private hospitals in 2021 (N = 406).**

| variable | category | frequency | percent |
|---|---|---|---|
| Knowledge of EMR | poor knowledge | 94 | 23.2 |
| | good knowledge | 312 | 76.8 |
| Computer skills on EMR | poor skill | 149 | 36.7 |
| | good skill | 257 | 63.3 |
| Language difficulty in using EMR | Yes | 145 | 35.7 |
| | no | 261 | 64.3 |
| Training on EMR | Yes | 216 | 53.2 |
| | no | 190 | 46.8 |
| Reason for not taking EMR training* | No have time to take training | 57 | 30.0 |
| | No access to take training | 105 | 55.6 |
| | Not interested to take training | 58 | 30.5 |
| | My work does not need training. | 35 | 18.4 |
| Using a computer for EMR | Yes | 245 | 60.3 |
| | no | 161 | 39.7 |
| Reason for using a computer for EMR* | Report writing | 82 | 33.5 |
| | Keeping patient file | 41 | 16.7 |
| | Listening music | 73 | 29.8 |
| | Using internet | 132 | 53.9 |
| | Reading | 77 | 31.4 |
| | Others | 46 | 18.8 |

* More than one answer is possible

adequate budget was allocated for the EMR system, and 168 (41.4%) of respondents received supportive supervision (M&E) for the EMR system (Table 3).

## Willingness to use the EMR system

Of all study participants, 307 (75.6%), 95% CI (71.4–79.8) expressed a willingness to use the EMR system in healthcare facilities. Approximately 64.3% and 74.4% of health professionals,

**Table 3. Technical factors for willingness to use an EMR system among health professionals working in Amhara region private hospitals in 2021 (N = 406).**

| variable | category | frequency | percent |
|---|---|---|---|
| Full computer access (desktop and laptop) | Yes | 230 | 56.7 |
| | no | 176 | 43.3 |
| Local area internet access | Yes | 183 | 45.1 |
| | no | 223 | 54.9 |
| Having technical support personnel | Yes | 212 | 52.2 |
| | no | 194 | 47.8 |
| Having an EMR implementation guideline | Yes | 225 | 55.4 |
| | no | 181 | 44.6 |
| Having budget allocations for EMR | Yes | 158 | 38.9 |
| | no | 248 | 61.1 |
| Having supportive supervision(M&E) | Yes | 168 | 41.4 |
| | no | 238 | 58.6 |
| Having management support | Yes | 210 | 49.0 |
| | no | 196 | 51.0 |

**Table 4. Willingness to use an EMR system among health professionals working in Amhara region private hospitals in 2021 (N = 406).**

| variable | category | frequency | percent |
|---|---|---|---|
| willingness to undergo computer training in order to use EMR | No | 104 | 25.6 |
| | yes | 302 | 74.4 |
| willingness to purchase a personal computer and to familiarize with the use of EMR | No | 145 | 35.7 |
| | yes | 261 | 64.3 |
| willingness to undergo training on EMR and its implementation | No | 116 | 28.6 |
| | yes | 290 | 71.4 |
| willingness to use EMR if properly trained | No | 122 | 30.0 |
| | yes | 284 | 70.0 |
| willingness to use EMR if the technical infrastructure is available | No | 127 | 31.3 |
| | yes | 279 | 68.7 |
| Overall willingness to use the EMR system | Not willing | 99 | 24.4 |
| | willing | 307 | 75.6 |

respectively, were willing to purchase and use a personal computer to familiarize themselves with EMR usage and to undergo computer training to enable EMR usage, and 70.0% were willing to use EMR if properly trained (Table 4).

## Factors associated with willingness to use the EMR system

Bivariable and multivariable binary logistic regression analyses were done to determine the association between the willingness to use the EMR system and independent variables. Accordingly, those variables that had a p-value of less than 0.2 in the bivariable regression analysis (educational level, knowledge of EMR, computer skill on EMR, language difficulty in using EMR, training on EMR, using a computer for EMR, full computer access, having technical support personnel, having supportive supervision (M&E), age group, having management support, and monthly income) were tested at the individual level, holding other variables constant against the dependent variable, and considered for the multivariable regression analysis. In the final multivariable logistic regression model, only the variables knowledge on EMR (AOR = 1.85, 95% CI (1.004–3.409)), training on EMR(AOR = 3.29, 95% CI (1.353–8.003)), having technical support personnel (AOR = 1.92, 95% CI (1.122–3.305)), supportive supervision (M&E) (AOR = 1.97, 95% CI (1.072–3.628)), and computer skill on EMR (AOR = 1.77, 95% CI (1.002–3.148)) were significant at p-value <0.05 with 95% confidence interval.

As shown in the Table 5 healthcare professionals who had good knowledge of EMR system software were 1.85 times more likely to be willing to use the EMR system than those who had poor knowledge of EMR system software (AOR = 1.85, 95% CI (1.004–3.409)); study subjects who had computer skills were 1.77 times more likely to have the willingness to use the EMR system compared to those with no have computer skills (AOR = 1.77, 95% CI (1.002–3.148)). The study also indicated that health professionals who had gotten an EMR training course were 3.29 times more likely to have the willingness to use the EMR system compared to those who had not got EMR training (AOR = 3.29, 95% CI (1.353–8.003)). Health professionals who had gotten technical support from technical support personnel were 1.92 times more likely to have a willingness to use the EMR system compared to those who had not gotten technical support (AOR = 1.92, 95% CI (1.122–3.305)). Respondents working in the presence of supportive supervision (M&E) were 1.97 times more likely to be willing to use the EMR system than their counterparts (AOR = 1.97, 95% CI (1.072–3.628)) (Table 5).

**Table 5. Factors associated with willingness to use EMR among health professionals working in Amhara region private hospitals in 2021 (N = 406).**

| variable | category | willingness | | Crud OR (95%, CI) | adjusted OR (95%, CI) |
|---|---|---|---|---|---|
| | | Willing | Not Willing | | |
| | | No (%) | No (%) | | |
| Educational level | Diploma | 97(23.9) | 45(11.1) | 1 | 1 |
| | Degree | 165(40.6) | 41(10.1) | 1.86(1.142–3.053) | 1.81 (0.969–3.391) |
| | Master & above | 45(11.1) | 13(3.2) | 1.60(0.789–3.270) | 1.18 (0.395–3.541) |
| Knowledge of EMR | poor knowledge | 56(13.8) | 38(9.4) | 1 | 1 |
| | good knowledge | 251(61.8) | 61(15.0) | 2.79(1.697–4.594) | 1.85 (1.004–3.409) * |
| Computer skill on EMR | poor skill | 93(22.9) | 56(13.8) | 1 | 1 |
| | good skill | 214(52.7) | 43(10.6) | 2.99(1.880–4.776) | 1.77 (1.002–3.148) * |
| Language difficulty on EMR | Yes | 99(24.4) | 46(11.3) | 1 | 1 |
| | no | 208(51.2) | 53(13.1) | 1.82(1.149–2.894) | 1.67 (.990–2.833) |
| Training on EMR | Yes | 185(45.6) | 31(7.6) | 3.32(2.053–5.388) | 3.29(1.353–8.003) ** |
| | no | 122(30.0) | 68(16.7) | 1 | 1 |
| Using a computer for EMR | Yes | 198(48.8) | 47(11.6) | 2.01(1.271–3.179) | 1.09 (0.602–1.973) |
| | no | 109(26.8) | 52(12.8) | 1 | 1 |
| Full (desktop & laptop) computer access | Yes | 192(47.3) | 38(9.4) | 2.68(1.681–4.272) | 0.69 (0.280–1.724) |
| | no | 115(28.3) | 61(15.0) | 1 | 1 |
| Having a technical support person | Yes | 176(43.3) | 36(8.9) | 2.35(1.473–3.754) | 1.92 (1.122–3.305) * |
| | no | 131(32.3) | 63(15.5) | 1 | 1 |
| Having supportive supervision(M&E) | Yes | 140(34.5) | 28(6.9) | 2.12(1.300–3.476) | 1.97(1.072–3.628) * |
| | no | 167(41.1) | 71(17.5) | 1 | 1 |
| Age group | 20–29 | 164(40.4) | 45(11.1) | 1 | 1 |
| | 30–39 | 121(29.8) | 43(10.6) | 0.77(0.478–1.247) | 0.72 (0.420–1.247) |
| | Above 40 | 22(5.4) | 11(2.7) | 0.54(0.248–1.216) | 0.95 (0.348–2.609) |
| Having management support | Yes | 156(38.4) | 43(10.6) | 1.34(0.853–2.123) | 0.79 (0.448–1.402) |
| | no | 151(37.2) | 56(13.8) | 1 | 1 |
| Monthly income | < 3200 | 18(4.4) | 4(1.0) | 0.68(0.162–2.863) | 0.34 (0.099–1.216) |
| | 3201–5250 | 101(24.9) | 44(10.8) | 0.34(0.127–0.950) | 0.31 (0.087–1.141) |
| | 5251–7800 | 118(29.1) | 32(7.9) | 0.55(0.202–1.547) | 0.22 (0.051–1.009) |
| | 7801–10900 | 37(9.1) | 14(3.4) | 0.40(0.130–1.232) | 0.55 (0.095–3.214) |
| | >10901 | 33(8.1) | 5(1.2) | 1 | 1 |

* = P value less than 0.05

** = P value less than 0.01

## Discussion

This institution-based cross-sectional survey was conducted to assess the health professionals' willingness to use EMR and to identify factors affecting willingness to use EMR in private hospitals in the Amhara region.

This study's respondents confirmed a good level of willingness to use EMR system. According to these findings, 75.6% health professionals were willing to use the EMR system in private hospitals. This finding is in line with a study carried out in private hospitals in India, where 75% were comfortable working on EMRs [10]. However, the result of this study is lower than that of a study conducted in Nigeria (Lagos) [15] and Kenya [16]. The plausible explanation for this difference could lead to inconsistencies in the inclusion of study subjects in the studies. Health professionals included in the current study were considered from all levels of private hospitals in the Amhara region, while respondents in Nigeria were taken only from tertiary

levels of hospital settings, and respondents from Kenya were taken from both private and governmental hospitals. Hence, the level of infrastructure, resource allocation, managerial support, and skilled manpower also varied across the types of health facilities, which makes a difference in utilizing the EMR system.

Another study done in Ethiopian health facilities proved that 85.9% of the experts were willing to use the EMR system [8]. when compared to this study, ours is slightly lower. The possible reason for this slight difference might be the variability of the study sample size. in the current study the size of study subject were less than when compared with the previous study. The other probable reason for this inconsistency could be the difference in the study setting, the setting of the previous study done was Bahirdar health facilities, those health facilities are selected based on their prior accessibility of EMR, but our study setting was all private hospitals in the Amhara region, this could affect the willingness of health professionals, since the availability of technical support and infrastructure, training, health workers technology resistance, and administrative problems varied across this health facility that makes a difference [4, 17].

In this study, health professionals who had good knowledge of the EMR systems were more likely to be willing to use the EMR system as compared to those with poor knowledge of the EMR system. This may be because health professionals with good knowledge tend to accept the advantages of technology and are more likely to be willing to use the EMR system. As discussed in other studies [8, 18, 19], health professionals who have good knowledge of the EMR system are more likely to be willing to use the EMR system than their counterparts or those who have poor knowledge of the EMR system. This is also a good explanation for the need to create awareness and strengthen continuous capacity building among less knowledgeable health professionals to narrow the knowledge gaps found about the EMR systems so that they will have a good willingness to use EMR and develop their knowledge for better use of the EMR system.

Despite differences among study participants, experience and availability of computerized EMR training were significantly associated. Health professionals who had trained in EMR systems were more likely to be willing to use the EMR system as compared to those who had not trained in any EMR systems before, and this is in line with other findings [8, 18, 20–22], which are explained by the evidence that training and education usually change people's views, willingness, knowledge, and skills on EMR systems [15].

Because computer skills are the foundation of information communication and EMR utilization in the healthcare system, more than half (63.3%) of the respondents in the current study had strong computer application skills. Other studies have indicated that there is a relationship between the level of IT skills and the willingness to use electronic medical records, and this study was no exception. Respondents with good computer skills were more likely than those with poor computer skills to be willing to use the EMR system. This finding was in line with the study findings from Ethiopia [8, 19] and Nigeria [15, 23]. This is because those health professionals with good computer skills and the availability of computers had a direct influence on those professionals' views on computer-based system use. The similarity of being a good skill was also likely explained by the availability of adequate computers, other resources, training centers, and support from various organizations.

Reliable and timely health information is at the heart of health system action, and information and communication technology initiatives such as EMRs help to improve the decision-making process. However, it is sometimes not available when required because of poor supportive supervision (M&E). However, in none of our literatures, supportive supervision was significantly associated with willingness to use EMR system. This study truly explained the fact that supportive supervision was an independent determinant for willingness to use the EMR system. This is because technical support supervision involves identifying or equipping a

relatively smaller team of technical staff with the relevant skillset, the right tools, and resources Thus, for the private as well as public health sectors to enhance the effectiveness and efficiency of supportive supervision, the technical support supervision approach should be adopted with a major focus on identifying the skill gaps among the technical staff at all levels, equipping the technical staff with the relevant skillset, motivating and retaining technical staff at all levels of health care, and facilitating the technical teams with the right tools and resources [24]. For this reason, about 52.2% of the respondents in the current study were working in the presence of technical support personnel, and respondents working in the presence of technical support personnel were more likely to be willing to use EMR systems than their counterparts were. This finding is supported by the other Ethiopian study [25], as well as the Saudi Arabia study [26]. Probably this will happen as long as individuals have access to technical support; their willingness to use a computer would increase because technical support would increase their technical skills, and which will lead to a greater willingness to use the EMR system. To alleviate this problem, health professionals should receive adequate training and technical support.

## Conclusion

The majority of healthcare professionals expressed a better willingness to use the EMR system 75.6%. The most significant factors associated with health professionals' willingness to use EMR systems were a lack of computer skills, a lack of computer training, and a lack of knowledge of EMR systems.

Therefore, healthcare organizations and stakeholders should improve the computer technical skills and EMR knowledge of health professionals by giving EMR training, providing ongoing follow-up, and offering technical assistance for health professionals.

## Supporting information

**S1 Data.**
(SAV)

**S1 Questionnaire. Informed consent of statement.**
(DOCX)

## Acknowledgments

The authors would like to express their deepest gratitude to the University of Gondar Institute of Public Health for the approval of ethical clearance. Next to this, I want to thank to all data collectors, supervisors, and respondents.

## Author Contributions

**Conceptualization:** Andualem Fentahun Senishaw, Biniyam Chakilu Tilahun, Araya Mesfin Nigatu, Shegaw Anagaw Mengiste, Karen Standal.

**Data curation:** Andualem Fentahun Senishaw.

**Formal analysis:** Andualem Fentahun Senishaw, Biniyam Chakilu Tilahun, Araya Mesfin Nigatu, Shegaw Anagaw Mengiste, Karen Standal.

**Funding acquisition:** Andualem Fentahun Senishaw.

**Investigation:** Andualem Fentahun Senishaw.

**Methodology:** Andualem Fentahun Senishaw.

**Project administration:** Andualem Fentahun Senishaw.

**Resources:** Andualem Fentahun Senishaw.

**Software:** Andualem Fentahun Senishaw.

**Supervision:** Andualem Fentahun Senishaw, Biniyam Chakilu Tilahun, Araya Mesfin Nigatu, Shegaw Anagaw Mengiste, Karen Standal.

**Validation:** Andualem Fentahun Senishaw.

**Visualization:** Andualem Fentahun Senishaw.

**Writing – original draft:** Andualem Fentahun Senishaw, Biniyam Chakilu Tilahun, Araya Mesfin Nigatu, Shegaw Anagaw Mengiste, Karen Standal.

**Writing – review & editing:** Andualem Fentahun Senishaw, Biniyam Chakilu Tilahun, Araya Mesfin Nigatu, Shegaw Anagaw Mengiste, Karen Standal.

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
