## [Decision Letter · Decision Letter 0]

13 Sep 2022

PONE-D-22-11490

Willingness to Use Electronic Medical Record (EMR) System and its Associated Factors among Health Professionals Working in Amhara region Private Hospitals 2021, Ethiopia

PLOS ONE

Dear Dr. Senishaw,

Thank you for submitting your manuscript to PLOS ONE. After careful consideration, we feel that it has merit but does not fully meet PLOS ONE’s publication criteria as it currently stands. Therefore, we invite you to submit a revised version of the manuscript that addresses the points raised during the review process.

Your article is interesting for both the reviewers, but one of them raised critical concerns about your research which need to be carefully addressed.

Below you can find other comments about the results of this round of review. 

We look forward to receiving your revised manuscript.

Kind regards,

Anna Prenestini, Ph.D.

Academic Editor

PLOS ONE

“The authors would like to forward the deepest gratitude to the University of Gondar institute of public health for the approval of ethical clearance and financial support for data collection. Next to this, thanks to all data collectors, supervisors, and respondents.”

3. Please amend the manuscript submission data (via Edit Submission) to include all authors' names.

Additional Editor Comments:

Thanks for your submission to PLOS ONE.

Your article is interesting for both the reviewers, but one of them raised critical concerns about your research.

In fact, the reviewer has previously published similar articles on the topic. Therefore I suggest to carefully read the comments and try to respond point by point.

In particular: which are the main difference between your study and the previous one of the reviewer? why did you not cite the already published articles on this topic and in the same context?

At least, the paper requires a deeper identification of the literature and its gaps.

Please take carefully into account the suggestions of the two reviewers and revise the paper.

Reviewers' comments:

Reviewer's Responses to Questions

**Comments to the Author**

1. Is the manuscript technically sound, and do the data support the conclusions?

Reviewer #1: No

Reviewer #2: Yes

2. Has the statistical analysis been performed appropriately and rigorously? 

Reviewer #1: Yes

Reviewer #2: Yes

3. Have the authors made all data underlying the findings in their manuscript fully available?

Reviewer #1: No

Reviewer #2: Yes

4. Is the manuscript presented in an intelligible fashion and written in standard English?

Reviewer #1: No

Reviewer #2: Yes

5. Review Comments to the Author

Reviewer #1: Abstract

1. Never use the abbreviation on the abstract…

2. How was this statement correct” Descriptive statistics with Binary logistic regression was performed to estimate the crude and adjusted odds ratio with a 95% Confidence interval? How descriptive statistics was regressed for logistic regression.

3. Do not begin with number

4. The conclusion is not SMART which is a carbon copy of the results without any change. In my view better to re-construct the conclusion in a manner that addresses the research question.

5. At conclusion, Why to say high, what is your ground to say as such?

Background

6. There are a # of editorial issues which needs English language subject expert editing

7. It is too long please shorten it focusing on :-

i. What is known on HIT & EMR?

ii. What is unknown about HIT & EMR, that you want to address by conducting this study?

iii. What are the major gaps in EMR in Ethiopia particularly in the private sector?

8. I have-not seen anything about magnitude & coverage of EMR in private sectors & what about the important factors influencing it?

9. What is the status of EMR implementation in private sectors...?

Methods

10. Why took 50% of willing to use EMR as a proportion; why not search for actual proportions, why not use my articles??? As a result I did not believe that it was representative. What bout for the second objective…?

11. Do you think that 13 Question is enough to assess level of knowledge...?

12. What about the cluster variability, do you know the facilities in Amhara region is with homogeneous characteristics...?

13. What makes this study different from studies conducted by myself and Dr. Mulusew Andualem’s article..? You need to justify how it differed from?

14. What does it mean by technical factor mean...? I think needs to be operationalized. Then; I think to technique mean sth related to skill. If so how can we related & categorize for knowledge technical factors... despite it might a prerequisite for skill that someone will develop.

15. I suspect a kind of plagiarism & what is important to repeat the same study within two years in the same population/area, I think wasting of resources.

16.

17.

18.

19.

Reviewer #2: This Topic will attract wide readership.Its original and contributes to knowledge. .

6. PLOS authors have the option to publish the peer review history of their article (what does this mean?). If published, this will include your full peer review and any attached files.

Reviewer #1: No

Reviewer #2: **Yes: **Rogers Shitiavai Songole

---

## [Author Response · Author response to Decision Letter 0]

5 Oct 2022

Here is a point-by-point response to the reviewers’ comments and concerns.

Comments from reviewer 1 

Comment 1: Never use the abbreviation on the abstract…

Response: Dear reviewers, Thank you for pointing out such important suggestions and valid comments, based on your comments we amend and exclude abbreviations in abstract part of revised manuscripts like EMR=electronic medical record, but other abbreviations like CI, AOR and SPSS are common and known abbreviations. 

Comment 2: How was this statement correct” Descriptive statistics with Binary logistic regression was performed to estimate the crude and adjusted odds ratio with a 95% Confidence interval? How descriptive statistics was regressed for logistic regression.

Response: dear reviewers, thank you for reminding us to revise this section. We made the revision on abstract part of revised manuscript based on your concern, and we changed “descriptive statistics with logistic regression” in to “descriptive statistics and logistic regression” to make it clear from abstract part in method section of revised manuscript line number 5.

Comment 3: Do not begin with number

Response: dear reviewers, thank you for reminding us to revise this section, we made the revision on abstract part of revised manuscript and highlighted the change.

Comment 4: The conclusion is not SMART, which is a carbon copy of the results without any change. In my view better to re-construct the conclusion in a manner, that addresses the research question.

Response: Dear Reviewers, We completely accepted your suggestion and comments; we revised and reconstructed the smart conclusion based on your concerns by drawing conclusion appropriately based on the data presented and based on its objective.

Comment 5: At conclusion, Why to say high, what is your ground to say as such?

Response: dear reviewers, Thank you for raising such important question, our ground to say high proportion was the finding result (75.6), and the result is above the half, but now we modified the term high in to good and incorporated it in the abstract part of conclusion section of revised manuscript.

Comment 6: There are a # of editorial issues, which needs English language subject expert editing 

Response: dear reviewers, Thank you for the great suggestion. We completely accepted your suggestion, and we revised entire manuscript editorial issues and English language edition with experts as much as possible to increase the readability of our manuscript. 

Comment 7: It is too long please shorten it focusing on:-

Response: dear reviewer, Thank you for this suggestion, based on your comment (i,ii,iii) we have revised the manuscript by shortening, and your suggestion is included under the introduction section of revised manuscript.

Comment 7.i: What is known on HIT & EMR?

Response: dear reviewers, Thank you for suggesting this important suggestions, we incorporated your suggestion by clearly stating what is known in EMR and HIT (i.e. in our sense HIT means HIS) in the introduction section Line number 3-5, 28-34, and 52-55.

Comment 2.ii: What is unknown about HIT & EMR, that you want to address by conducting this study?

Response: dear revisers, Thank you again for this suggestion, based on your comment we have revised the manuscript by clearly stating what is unknown on EMR, and your suggestion is included under the introduction section line number 20-25 and 36-44.

Comment 7.iii: What are the major gaps in EMR in Ethiopia particularly in the private sector?

Response: dear reviewers, we completely accepted your suggestion. We revised and added major gaps of EMR in private sectors to the revised manuscript by including your suggestions under the introduction section line number 30-35

Comment 8: I have-not seen anything about magnitude & coverage of EMR in private sectors & what about the important factors influencing it?

Response: dear reviewers, Thank you for giving your time to review our paper. Even though the magnitude and coverage of EMR in private sector is not widely available in published literatures, we tried to include your comment by founding different data on the influencing factors and coverage of EMR in private sectors under the introduction section line number 11-18. 

Comment 9: What is the status of EMR implementation in private sectors...?

Response: dear reviewers: Thank you for raising such important question. Of course, we stated the status of EMR implementation in private sector a little, even though the status EMR implementation in Ethiopian private sector is not known in scientific way, we got the status through asking the private hospitals and we included it in introduction section of revised manuscript line number 12-15. 

Comment 10: Why took 50% of willing to use EMR as a proportion; why not search for actual proportions, why not use my articles??? As a result, I did not believe that it was representative. What bout for the second objective…?

Response: dear reviewers, Thank you so much for valuable comments what you raised on taking 50% proportion, we have used 50% proportion for willingness to use EMR in private hospital because there is no study conducted related with willingness to use EMR in Ethiopia. Sorry, we are not able to distinguish which one your article is? If it is mr, Birhanu`s Article conducted in Bahirdar city, this article is not conducted necessarily on private hospital rather it is conducted in Dahrrdar public health facilities.

For second objective, there is no other study conducted in Ethiopia related with second objective in private hospitals.

Comment 11: Do you think that 13 Question is enough to assess level of knowledge...?

Response: dear reviewers, thank you for raising the question unclear with you. Yes, 13 question is enough to assess level of knowledge because these questions was taken from other study and adapted in to our study by pre testing and validating with experts revision before data collection, and then by categorizing Likert scale value in to good knowledge and poor knowledge. Beside that reliability test were performed. 

Comment 12: What about the cluster variability, do you know the facilities in Amhara region is with homogeneous characteristics...?

Response: dear reviewers, thank you for your important and constructive comment. Of course you may think the cluster variability in different facilities, But the study setting was one region, the characteristics of hospitals in one region has homogenous characteristics, since the level of facility is the same hospitals, with the same geographical area in Ethiopia. If it was different level of health facility like health center, clinics and hospitals, it may happen. 

Commnet 13: What makes this study different from studies conducted by myself and Dr. Mulusew Andualem’s article..? You need to justify how it differed from?

Response: dear reviewers, thank you for you important questions. The study conducted by Dr. mulusew Andualem, and you is different from me by:

In terms of ownership: your study is public or not all health facility you studied owned by private, whereas this study is only private owned or private hospitals.

In terms of facility: your study is in different health facility including clinic, health centers and hospitals, whereas this study is conducted in hospitals only

In terms of setting: your study is only in bahirdar city, where as this study is including all cities in Amhara region having private hospitals (4 city)

In terms of sample size, sampling technique and source population, and this article is new and original since there is no study conducted in private hospitals in Ethiopia about willingness to use EMR.

Comment 14: What does it mean by technical factor mean...? I think needs to be operationalized. Then; I think to technique mean sth related to skill. If so how can we related & categorize for knowledge technical factors... despite it might a prerequisite for skill that someone will develop.

Response: dear reviewers, Thank you for raising such important question and also thanks for reminding us to operationalize the term technical factors, dear reviewer based on your suggestion we operationalized technical factors, but when we say technical factor, it is general term, which can incorporate computer related terms in addition to skills under the method section.

Dear reviewer, thank you very much for giving comments on how can to categorize knowledge on technical factors without operationalizing it, even though during proposal development from conceptual diagram we categorize knowledge under technical factor without operationalizing, now we operationalize technical factors as it includes knowledge in addition to other technical related factors. 

Comment 15: I suspect a kind of plagiarism & what is important to repeat the same study within two years in the same population/area, I think wasting of resources.

Response: dear reviewers, Thank you for raising your idea on plagiarism and on being the same study. However, our study is original and novel on private sector, since private sectors health structure is different from government owned. Even though the population is health professional, they are in private hospitals and study area is different from your studies. For your suspecting of plagiarism, we never take direct copy of another idea, rather we support another idea by citing properly. 

Comments from reviewer 2 

Reviewer #2: “This Topic will attract wide readership. Its original and contributes to knowledge.”

Response: dear reviewers, Thank you very much for your appreciation and your recognition of our study.

---

## [Decision Letter · Decision Letter 1]

27 Dec 2022

PONE-D-22-11490R1Willingness to Use Electronic Medical Record (EMR) System and its Associated Factors among Health Professionals Working in Amhara region Private Hospitals 2021, EthiopiaPLOS ONE

Dear Dr. Senishaw,

Thank you for submitting your manuscript to PLOS ONE. After careful consideration, we feel that it has merit but does not fully meet PLOS ONE’s publication criteria as it currently stands. Therefore, we invite you to submit a revised version of the manuscript that addresses the points raised during the review process.

Please find below the comments on your paper. 

We look forward to receiving your revised manuscript.

Kind regards,

Anna Prenestini, Ph.D.

Academic Editor

PLOS ONE

Additional Editor Comments:

Dear authors,

please, find here attached you can find the response of reviewer #1.

Reviewer #2 sent his/her minor revision to the original submission, so I decided not to ask another reviewer for the second review in this second round.

Nevertheless, reviewer #1 raises other concerns about your response and improvements after the first round. Therefore, I recommend following the comments in order to give a more appropriate response to his/her concerns on the paper.

Especially when he/she raises problems already expressed in the original submission.

Moreover, sending the manuscript to a professional proofreading service is necessary to improve the English language.

Please, send a point-by-point letter with your response to the comments raised by the reviewer.

Hope to read the new version of the paper soon.

Kind regards.

Reviewers' comments:

Reviewer's Responses to Questions

**Comments to the Author**

1. If the authors have adequately addressed your comments raised in a previous round of review and you feel that this manuscript is now acceptable for publication, you may indicate that here to bypass the “Comments to the Author” section, enter your conflict of interest statement in the “Confidential to Editor” section, and submit your "Accept" recommendation.

Reviewer #1: All comments have been addressed

2. Is the manuscript technically sound, and do the data support the conclusions?

Reviewer #1: Partly

3. Has the statistical analysis been performed appropriately and rigorously? 

Reviewer #1: Yes

4. Have the authors made all data underlying the findings in their manuscript fully available?

Reviewer #1: Yes

5. Is the manuscript presented in an intelligible fashion and written in standard English?

Reviewer #1: No

6. Review Comments to the Author

Reviewer #1: 1. Still the conclusion needs to be revised in a manner that the take of massage is boldly communicated.

2. The queries related the issue of sampling is not yet addressed.

3. The issues of plagiarism is not narrating in away that is convincing.

4. There is problems of English language editorials.

5. Questions # 12 & 14 has to be seen carefully & addressed.

7. PLOS authors have the option to publish the peer review history of their article (what does this mean?). If published, this will include your full peer review and any attached files.

Reviewer #1: No

---

## [Author Response · Author response to Decision Letter 1]

13 Jan 2023

dear reviewer and editors, we highly appreciate and acknowledge for your time to review our paper, thank you very much

---

## [Editor Report · Decision Letter 2]

7 Feb 2023

Willingness to Use Electronic Medical Record (EMR) System and its Associated Factors among Health Professionals Working in Amhara region Private Hospitals 2021, Ethiopia

PONE-D-22-11490R2

Dear Dr. Senishaw,

We’re pleased to inform you that your manuscript has been judged scientifically suitable for publication and will be formally accepted for publication once it meets all outstanding technical requirements.

Kind regards,

Anna Prenestini, Ph.D.

Academic Editor

PLOS ONE
---

## [Editor Report · Acceptance letter]

19 Apr 2023

PONE-D-22-11490R2 

Willingness to Use Electronic Medical Record (EMR) System and its Associated Factors among Health Professionals Working in Amhara region Private Hospitals 2021, Ethiopia 

Dear Dr. Senishaw:

I'm pleased to inform you that your manuscript has been deemed suitable for publication in PLOS ONE. Congratulations! Your manuscript is now with our production department. 

Kind regards, 

on behalf of

Professor Anna Prenestini 

Academic Editor

PLOS ONE